# Optimization of a Method for Detecting Intracellular Sulfane Sulfur Levels and Evaluation of Reagents That Affect the Levels in *Escherichia coli*

**DOI:** 10.3390/antiox11071292

**Published:** 2022-06-29

**Authors:** Qiaoli Yu, Mingxue Ran, Yuqing Yang, Huaiwei Liu, Luying Xun, Yongzhen Xia

**Affiliations:** 1State Key Laboratory of Microbial Technology, Shandong University, 72 Binhai Road, Qingdao 266237, China; yuqiaoli@mail.sdu.edu.cn (Q.Y.); ranmx@sdu.edu.cn (M.R.); angyuq@mail.sdu.edu.cn (Y.Y.); liuhuaiwei@email.sdu.edu.cn (H.L.); 2School of Molecular Biosciences, Washington State University, Pullman, WA 99164-7520, USA

**Keywords:** intracellular sulfane sulfur, thiosulfate, redox homeostasis, sulfur-bearing compounds, *Escherichia coli*

## Abstract

Sulfane sulfur is a class of compounds containing zero-valent sulfur. Most sulfane sulfur compounds are reactive and play important signaling roles. Key enzymes involved in the production and metabolism of sulfane sulfur have been characterized; however, little is known about how to change intracellular sulfane sulfur (iSS) levels. To accurately measure iSS, we optimized a previously reported method, in which reactive iSS reacts with sulfite to produce thiosulfate, a stable sulfane sulfur compound, before detection. With the improved method, several factors were tested to influence iSS in *Escherichia coli*. Temperature, pH, and osmotic pressure showed little effect. At commonly used concentrations, most tested oxidants, including hydrogen peroxide, tert-butyl hydroperoxide, hypochlorous acid, and diamide, did not affect iSS, but carbonyl cyanide m-chlorophenyl hydrazone increased iSS. For reductants, 10 mM dithiothreitol significantly decreased iSS, but *tris*(2-carboxyethyl)phosphine did not. Among different sulfur-bearing compounds, NaHS, cysteine, S_2_O_3_^2−^ and diallyl disulfide increased iSS, of which only S_2_O_3_^2−^ did not inhibit *E. coli* growth at 10 mM or less. Thus, with the improved method, we have identified reagents that may be used to change iSS in *E. coli* and other organisms, providing tools to further study the physiological functions of iSS.

## 1. Introduction

Sulfide (H_2_S and HS^−^) is considered the third gaso-transmitter in mammals, participating in various physiological functions [1,2]. Recent reports show that sulfide signaling is usually via intracellular sulfane sulfur (iSS) [3,4]. Sulfane sulfur, containing zero-valence sulfur, comes in several forms, such as inorganic and organic polysulfide (HS_n_^−^, RS_n_^−^, and RSS_n_R; n ≥ 2) and elemental sulfur [5]. It modifies protein cysteine (Cys) thiols to form persulfide (*S*-sulfhydration), which alters protein configurations and amends the catalytic or regulatory activities [6]. Glyceraldehyde-3-phosphate dehydrogenases in *Escherichia coli* and *Staphylococcus aureus* are inhibited after their active site Cys residues are *S*-sulfhydrated [7,8], but the enzyme activity from the mouse liver is enhanced after *S*-sulfhydration [9]. Several bacterial gene regulators have been identified to respond to iSS. MgrA senses iSS after sulfide-stress to activate the expression of virulent factors in *S. aureus* [8]. The iSS level in *Pseudomonas aeruginosa* is high in the late log phase and early stationary phase of growth, and it activates MexR to turn on the expression of a multiple drug efflux pump MexAB when cells enter the stationary phase [10]. The high levels of iSS in the late log phase and early stationary phase of growth also significantly enhance the activity of the master quorum-sensing activator LasR in *P. aeruginosa* [11]. The multidrug repressor MarR is inactivated by increased iSS in the early stationary phase of growth of *E. coli* [12]. When iSS is high, it modifies the oxidative stress regulator OxyR that activates the expression of genes involved in lowing iSS in *E. coli* [13]. Hence, iSS in bacteria also plays a critical role in regulating different physiological processes.

The iSS is unstable, as it can be oxidized by oxygen, reduced by thiols, and decomposed under acid conditions [14,15]. Therefore, the growth conditions and oxidative and reducing agents may interfere with iSS in bacteria. The growth conditions include osmolarity, pH, temperature, and dissolved oxygen [16,17,18,19]. Reactive oxygen species (ROS) cause oxidative stress [20,21], damaging proteins, nucleic acids, and lipids [22]. Superoxide radicals (O_2_^•−^), hydrogen peroxide (H_2_O_2_), and hydroxyl radicals (•OH) are common ROS, and they are produced via the electron transport chain of aerobic respiration [23]. Reagents, such as hypochlorous acid and carbonyl cyanide m-chlorophenyl hydrazone (CCCP), are often used to induce oxidative stress in cells. Hypochlorous acid is a strong oxidant, and it reacts with DNA, protein, and lipids [24]. CCCP is an uncoupling agent that inhibits oxidative phosphorylation and promotes ROS production [25], which induces oxidative stress in cells [26]. The reagents diamide and tert-butyl hydroperoxide (tBH) are common thiol oxidants [27]. Commonly used reducing agents include dithiothreitol (DTT) and tris(2-carboxyethyl)phosphine (TCEP). How these factors influence iSS levels in bacteria is unclear.

The production of iSS also affects its homeostasis. The metabolism of Cys is the main source of sulfane sulfur in *Escherichia coli* [28]. Cys is either present in a rich medium or is produced from sulfate [29]. Sulfate is reduced via assimilatory sulfate reduction to sulfide before being incorporated into Cys. The yeast *Saccharomyces cerevisiae* can convert thiosulfate to iSS which is then reduced to sulfide for Cys biosynthesis [30]. Further, the yeast *Schizosaccharomyces pombe* directly uses sulfane sulfur to produce Cys [31]. Garlic oil is rich in sulfane sulfur-containing compounds with diallyl disulfide (DADS) and diallyl trisulfide being prevalent [32]. The supply of these sulfur compounds in the growth medium may affect iSS.

Recently, we have developed a simple and sensitive method to detect total iSS from biological samples. SO_3_^2−^ is used to react with unstable iSS to produce stable S_2_O_3_^2−^ under hot and alkaline conditions (pH = 9.5, 95 °C). S_2_O_3_^2^^−^ is then derived with mono-bromobimane (mBBr) to form bimane-S_2_O_3_^2−^, which is detected by using high-performance liquid chromatography (HPLC) with a fluorescence detector. The thiosulfate content of the samples incubated in the control buffer without sulfite is the blank sample that is subtracted from the test samples [33]. Thiosulfate is also a sulfane sulfur-containing compound [1], but it does not react with cellular thiols under physiological conditions [34,35]. This method was named as sulfite-dependent sulfane sulfur detection method (SdSS), and it is sensitive to detect active sulfane sulfur in bacteria, plants, and animals [10,31,33,36] as well as in wine [37].

However, after extensive use, we noticed several shortcomings that affect the accuracy and sensitivity of this method. The peaks of mBBr derivatized S_2_O_3_^2−^ and glutathione (GSH) partially overlapped, affecting the detection sensitivity of bimane-S_2_O_3_^2−^; the washing process of preparing the bacterium samples might cause iSS loss; a fraction of iSS was converted to thiosulfate without sulfite during heating. Here, we addressed these issues and revised the method. With this improved method, we investigated the factors that could affect iSS in *E. coli*. The key findings included that thiosulfate is a good reagent to increase iSS without affecting bacterial growth.

## 2. Materials and Methods

### 2.1. Bacterial Strains, Culture Conditions, and Reagents

*Escherichia coli* BL21(DE3) was grown in the lysogeny broth (LB) at 37 °C. Sodium sulfite, sodium thiosulfate, sodium sulfate, GSH, Cys, NaHS, DTT, mBBr, tBH, DADS, TCEP, diamide (CAS: 10465-78-8), and CCCP were purchased from Sigma-Aldrich (Burlington, MA, USA). Diethylenetriaminepentaacetic acid (DTPA), FeSO_4_·7H_2_O, acetic acid, sodium hypochlorite, and H_2_O_2_ were purchased from Bio Basic Inc (Markham, ON, Canada). TritonX-100 was purchased from Sangon Biotech (Shanghai, China). Rosup is a compound mixture of oxidative reagents, and it is used as a positive control in the Reactive Oxygen Species Assay Kit (Beyotime Biotech Inc., Shanghai, China).

### 2.2. Sample Preparations for iSS Detection

Colonies of *E. coli* BL21(DE3) were picked up and cultured in LB medium overnight. The cultures were transferred into a fresh medium at an initial OD_600_ of 0.05 for aerobic cultures or at an initial OD_600_ of 0.02 for anaerobic cultures. Cells were at defined time intervals, and the OD_600_ value was measured with the spectrophotometer UV1800 (Shimadzu, Kyoto, Japan). Three different procedures were used to harvest cells. Option 1 (No-wash): *E. coli* cells were transferred into a microfuge tube and harvested by centrifuging at 3300× *g* for 5 min at 4 °C. The supernatants were carefully removed with a pipettor, and the collected cells were used for iSS measurement. Option 2 (Double-centrifugation): after the cells were harvested and the supernatant was removed, the tubes with pellets were centrifuged again at 3300× *g* for 1 min. The residual supernatant was carefully removed with a pipettor. Option 3 (Wash-once): after the cells were harvested and the supernatant was removed, the collected pellets were washed once with 1 mL 100 mM Tris-HCl buffer (pH = 7.4) to remove the residual supernatant. Wash-once is the reported procedure [33]. No-wash and Double-centrifugation were two selected procedures for comparison. Double-centrifugation was found to be the optimal procedure and was adapted for iSS detection.

### 2.3. The Optimized iSS Detection Method

The previously reported SdSS method is designed to measure iSS [33]. The SdSS method was optimized. Briefly, the reaction buffer was prepared with 50 mM Tris-HCl buffer (pH = 9.5) containing 1% TritonX-100, 50 μM DTPA, and 1 mM sulfite, and the control buffer contained no sulfite but 0.5 mM DTT. DTT was introduced to reduce sulfate sulfur to H_2_S, avoiding the spontaneous oxidation of sulfane sulfur to thiosulfate during subsequent steps. One mL of cells at OD_600_ of 1.0 was harvested with double-centrifugation and resuspended in 100 μL of the reaction buffer or the control buffer. The resuspended cells were incubated at 95 °C for 10 min. The samples were then centrifuged at 15,700× *g* for 3 min; 50 μL of the supernatant was mixed with 5 μL of 25 mM mBBr and incubated in the dark at room temperature for 25 min to convert S_2_O_3_^2−^ into bimane-S_2_O_3_^2−^ adduct. 110 μL of an acetic acid and acetonitrile mixture (*v*/*v*, 1:9) was added to stop the reaction and denature proteins. The mixtures were centrifuged to precipitate cell debris and denatured proteins at 15,700× *g* for 3 min.

The bimane-S_2_O_3_^2−^ adduct in the supernatant was determined by using HPLC (LC-20A, Shimadzu, Kyoto, Japan) with a fluorescence detector (RF20A, Shimadzu, Kyoto, Japan). The gain value and sensitivity of this fluorescence detector were set as “medium” and “×16”, respectively. Briefly, 5 μL supernatant was injected onto a reverse-phase C18 column (VP-ODS, 150 × 4 mm, Shimadzu, Kyoto, Japan) with a guard column (Inertsil ODS-SP 5 μm 5020-19006, Shimadzu, Kyoto, Japan) through an autosampler (SIL-20A, Shimadzu, Kyoto, Japan). The column was maintained at 38 °C in a column thermostat (CTO-20A, Kyoto, Japan), and eluted with a gradient solution A (0.25% acetic acid and 10% methanol in distilled water, with pH being adjusted to 3.9 by using NaOH) and solution B (0.25% acetic acid and 90% methanol in distilled water) from 8% B to 40% B in 7 min, 40% B for 5 min, 40% B to 100% in 0.1 min, 100% B for 6 min at a flow rate of 0.8 mL/min. The adduct was detected by a fluorescence detector with an optimized excitation wavelength (Ex) and emission wavelength (Em) at 380 nm and 466 nm, respectively. The bimane-S_2_O_3_^2−^ adduct was normally detected at a retention time of 13.0 min.

### 2.4. The Effect of Growth Conditions and Reagents on E. coli iSS

The cells of *E. coli* BL21(DE3) were transferred into a fresh LB medium and incubated at 37 °C until OD_600_ reached about 1.0. The cultures were aliquoted and incubated under various conditions, including different temperatures, pH, and osmolarities, or spiked with sulfur-containing compounds (NaHS, sulfite, sulfate, Cys, thiosulfate, DADS, and GSH), oxidants (H_2_O_2_, tBH, CCCP, diamide, sodium hypochlorite, Rosup, and Fenton’s reagent) and reductants (DTT and TCEP). DADS and CCCP were dissolved in absolute ethanol and DMSO, respectively. The other reagents were dissolved in distilled water. Fenton reagent was prepared by mixing 10 mM H_2_O_2_ with 10 mM FeSO_4_·7H_2_O. The final concentration of each reagent was given in the results. After incubation for 30 min or as specified in the text [33], the cells were collected by using the double-centrifugation method and analyzed with our optimized SdSS method.

The iSS content in *E. coli* cells was also assayed with cells cultured under aerobic and anaerobic conditions. For aerobic growth, the cells were cultured in 50 mL of LB medium. For anaerobic growth, the cells were cultured in 100 mL serum bottles sealed with butyl rubber stoppers. 70 mL nitrogen deoxygenated LB medium was filled into the bottles before sterilization. The cells were inoculated in the bottles by using a syringe. Cells were taken at defined time intervals given in the results. The iSS content was detected by using our optimized method.

## 3. Results

The detection sensitivity for sulfane sulfur was improved by optimizing HPLC conditions.

Sulfane sulfur is converted to S_2_O_3_^2−^, which is derivatized with mBBr to bimane-S_2_O_3_^2−^ for detection [33]; however, the peaks of mBBr-derivatized S_2_O_3_^2−^ and GSH partially overlapped within the chromatogram (Figure 1A). The HPLC elution program was optimized to separate the two peaks (Figure 1B). The maximal Ex and Em of bimane-S_2_O_3_^2−^ were determined to be 380 nm and 466 nm, respectively (Figure 2A,B). When the maximal wavelengths were used, the peak area of bimane-S_2_O_3_^2−^ increased about 20.0% over that obtained with the reported Ex and Em (Figure 2C). Nine combinations of the settings of gain value and sensitivity of the fluorescence detector were tested to increase the signal-to-noise ratio (SNR) for the bimane-S_2_O_3_^2−^ adduct (Appendix A), and the highest value of the SNR was when the gain value was “×16” and the sensitivity was “medium”.

By using the optimized HPLC conditions, the standard curves of bimane-S_2_O_3_^2−^ ranging from 0.5 μM to 5 μM in 50 mM Tris-HCl buffer (pH = 9.5) with or without *E. coli* BL21(DE3) at OD_600_ of 1.0 was determined. With the cells, the baseline was increased, but the detection of added S_2_O_3_^2−^ was not affected (Figure 3A). Further, the detection limit of bimane-S_2_O_3_^2−^ in the Tris-HCl buffer was improved to 10 nM by using the optimized conditions (Figure 3B).

### 3.1. The Optimization of Bacterial Samples Preparation

*E. coli* iSS was tested by suspending the cells in the sample buffer (with SO_3_^2−^) and the control buffer (without SO_3_^2−^), and heating was used to convert iSS and SO_3_^2−^ to S_2_O_3_^2−^ [33]. Whether iSS was self-oxidized to S_2_O_3_^2−^ during the heating process was tested by using DTT to reduce iSS to H_2_S, as DTT readily reduces reactive sulfane sulfur, such as organic persulfide, to the corresponding thiol and H_2_S [38]. *E. coli* cells were treated in the control buffer. When 0.2 mM DTT or more was added into the control buffer, S_2_O_3_^2−^ in the control group decreased from 8.5 × 10^−2^ nmol/mL/OD to 1.5 × 10^−2^ nmol/mL/OD (Figure 4A). When S_2_O_3_^2−^ was added to the control buffer, DTT did not reduce it after heating (Figure 4B). Thus, 0.5 mM DTT was subsequently included in the control buffer without SO_3_^2−^ to prevent S_2_O_3_^2−^ formation from the autoxidation of iSS during the heating process.

Whether it is necessary to remove the residual culture supernatant before iSS determination was tested. Sulfane sulfur in the supernatant fluctuated around 23 µM without significant changes during the growth of *E. coli* BL21(DE3) in the LB medium (Appendix A). When the harvested cells by centrifugation were washed once with Tris-HCl buffer (50 mM, pH = 7.4) as reported [33], iSS was 226.5 (10^−3^ nmol/mL/OD) (Figure 5A). When the harvested cells were directly measured without washing, iSS was 460.9 (10^−3^·nmol/mL/OD) (Figure 5A). The unwashed cell pellet contained several microliters of the culture supernatant, which contributed to the increased iSS. The residual supernatant was removed by second centrifugation to collect the liquid on the wall of the microfuge tube and pipetting removal (double-centrifugation step). With the double-centrifugation step, iSS was 281.1 (10^−3^·nmol/mL/OD) (Figure 5A). This double-centrifugation step was adopted because it minimizes the culture supernatant and prevents the loss of iSS during washing.

Various volumes of an *E. coli* BL21(DE3) culture at OD_600_ were used to detect iSS contents. With the revised method, iSS concentrations could be accurately detected with 1 mL of *E. coli* BL21(DE3) cells at OD_600_ of 1.0, but the accuracy was reduced with sample volume smaller than 1 mL (Figure 5B). This optimized SdSS method was used in the following tests.

### 3.2. The Effects of Different Stress Factors on iSS Content of E. coli

When cell cultures of *E. coli* BL21(DE3) in LB medium were incubated at different temperatures for up to 30 min, the iSS contents were not significantly changed (Appendix A). When cells were resuspended in LB or Tris-HCl buffer at different pH values, the iSS content in *E. coli* cells was not significantly changed (Appendix A). 1–8% NaCl was added directly into the cultures to change the osmotic pressure, but it did not change the iSS content of cells, either (Appendix A).

To test the effect of oxygen on iSS, the iSS content in *E. coli* was tested when the bacterium was cultured at different growth phases under both aerobic and anaerobic conditions. The iSS content under aerobic conditions rose to the highest level rapidly in the mid-log phase and remained high till the early stationary phase. It significantly decreased during the stationary phase (Figure 6A). However, under anaerobic conditions, the iSS content gradually increased during the entire log phase and reached its maximum in the early stationary phase. The high level of iSS content was relatively stable during the stationary phase (Figure 6B).

### 3.3. Effects of Cellular Redox Balance on E. coli iSS

Different compounds that could disturb the intracellular redox potential were used to test if they could change the homeostasis of sulfane sulfur in cells after 30-min treatment. DTT, TCEP, tBH, H_2_O_2_, sodium hypochlorite, diamide, and Fenton’s reagent were tested at 1, 0.2, 0.25, 0.1, 0.1, 1, and 0.1 mM, respectively. CCCP and Rosup were used at 10 μM and 50 μg/mL. They are the reported concentrations in the literature and kit instruction [39,40,41,42,43,44,45,46]. Only CCCP and Rosup promoted iSS (Figure 7A). When tested at high concentrations, 10 mM Rosup, 10 mM diamide, and 5 mM Fenton’s reagent increased iSS, but 10 mM H_2_O_2_ and tBH did not (Appendix A). For reductants, 1 mM DTT and 0.2 mM TCEP did not significantly change iSS (Figure 7A). When tested at high concentrations, 10 mM DTT reduced iSS, but 10 mM TCEP did not (Appendix A).

### 3.4. Effects of Exogenous Sulfur-Bearing Compounds on E. coli iSS

Different sulfur-bearing compounds at 10 mM were added into *E. coli* BL21(DE3) cultures to test if they affected iSS. Cys, NaHS, DADS, and S_2_O_3_^2−^ increased the iSS content of *E. coli*, but GSH, SO_3_^2−^ and SO_4_^2−^ did not (Figure 7B). Low concentrations of S_2_O_3_^2−^, DADS, Cys, and NaHS were further tested, and they still increased iSS but at reduced magnitudes (Appendix A). S_2_O_3_^2−^ was still the most effective, and it significantly increased iSS even at 0.5 mM.

The sulfur-bearing compounds which could promote iSS content were added to BL21(DE3) culture to observe whether they were toxic to cells at different concentrations. In a closed environment, all of these sulfur compounds except S_2_O_3_^2−^ repressed cell growth of *E. coli* to different degrees (Appendix A). S_2_O_3_^2−^ at 10 mM or less did not show apparent inhibition. The DADS and NaHS showed more severe inhibition than Cys. In an open environment, the inhibition effect of DADS and NaHS was largely relieved (Appendix A), likely because H_2_S and DADS were volatile and evaporated. Again, S_2_O_3_^2−^ did not affect cell growth (Appendix A).

Nontoxic S_2_O_3_^2−^ at 2 mM was added into the cell cultures of *E. coli* in LB medium to observe the change of iSS during growth. The iSS content greatly increased in comparison with the control (Figure 8). The iSS content reached the highest level at the end of logarithmic growth. Then it gradually decreased to a level similar to that of the control group after 24 h of growth.

## 4. Discussion

The previously reported SdSS method for iSS detection in bacteria was optimized. DTT is a key factor for optimization. DTT is a common reducing agent, and it reduces sulfane sulfur to H_2_S [47]. The addition of DTT in the control buffer prevented the oxidation of iSS to thiosulfate, which interferes with the SdSS method. The inclusion of DTPA in both the reaction buffer and the control buffer is to chelate transition metals that catalyze sulfur oxidation [48]. Another improvement is the revised HPLC elution method. The revised method separates bimane-S_2_O_3_^2−^ and bimane-GS so that the high concentrations of bimane-GS derived from *E. coli* will not interfere with bimane-S_2_O_3_^2−^ detection (Figure 1B). Third, we optimized the fluorescence detection settings (Figure 1B and Figure 2C and Appendix A), and the detection threshold for bimane-S_2_O_3_^2−^ was improved from 200 nM to 10 nM (Figure 3B). Finally, the double-centrifugation method to prepare cells before iSS detection was recommended (Figure 5A), as it minimizes culture supernatant and prevents the loss of iSS during washing with a buffer that changes the culturing environment of the tested cells.

We had expected a decrease in iSS when ROS was added to cell suspensions, as sulfane sulfur is known to react with ROS [49,50,51]; however, several tested reagents that induce oxidative stress increased iSS (Figure 7A). A possible explanation is that intracellular acid-labile sulfur is also oxidized by the addition of these reagents. The acid-labile sulfur, including Fe-S clusters, is sulfide [52,53]. When O_2_^•−^ and HO• oxidize Fe-S clusters and release Fe^2+^ [54,55,56], the sulfur in the cluster is theoretically oxidized to sulfane sulfur (zero valences), as sulfide reacts with ROS to produce sulfane sulfur [57]. H_2_O_2_ and tBH were unable to increase iSS (Figure 7A), perhaps partly because they are rapidly metabolized by *E. coli* cells [58], partly because they react with sulfane sulfur at a relatively slow rate [40], and partly because they do not directly damage Fe-S clusters [59].

Sulfane sulfur, ROS, and Reactive chlorine species (RCS) are signaling molecules, and their signaling pathways may overlap [53,60,61]. OxyR is a major global regulator of *E. coli* in response to oxidative stress [62], and it also senses iSS through the persulfidation of Cys^199^ [13]. Other redox-based transcriptional regulators also used Cys residues for signal sensing [61,63]. Two members of the MarR (multiple drug-resistant regulators) families that repress multiple drug efflux pumps have been shown to use Cys residues to sense both H_2_O_2_ and iSS [10,12]. Further, our data show that ROS and RCS significantly increased iSS in *E. coli* cells (Figure 7A). The results imply that iSS may participate in the signaling transduction induced by ROS or RCS. Further studies are needed to understand whether sulfane sulfur is involved in the signaling induced by ROS and RCS.

S_2_O_3_^2−^ is the only tested sulfur donor that could increase the iSS content without affecting bacterial growth (Appendix A and Figure 8), and it may be used to increase iSS in *E. coli* or other organisms to evaluate the effect of elevated iSS on cells. Although S_2_O_3_^2−^ may be used with other organisms to increase iSS, its concentration should be tested. For example, *Saccharomyces cerevisiae* is partially inhibited by 10 mM S_2_O_3_^2−^, as high concentrations of S_2_O_3_^2−^ directly inhibit cytochrome c oxidase of the electron transport chain in the yeast mitochondria [35]. *E. coli* is not inhibited by 10 mM S_2_O_3_^2−^ (Figure 8), likely because it does not use cytochrome c oxidase in its electron transport chain [64]. There are two known metabolic pathways of S_2_O_3_^2−^ in *E. coli*: one is catalyzed by CysM that uses S_2_O_3_^2−^ to produce Cys [65,66], and Cys is then converted to sulfane sulfur [28]; the other is catalyzed by rhodanese (RHOD) GlpE that directly converts S_2_O_3_^2−^ to sulfane sulfur [65]. *E. coli* contains eight proteins carrying RHOD domains [67]. Further studies are necessary to identify whether CysM or one or more RHODs are responsible to convert S_2_O_3_^2−^ to sulfane sulfur.

In summary, we optimized the SdSS method and used *E. coli* as a model to extensively investigate the effects of different stress factors and reagents on iSS homeostasis. This work not only provides a better method for analyzing iSS in *E. coli* and possibly other biological samples but also investigated several factors possibly affecting iSS homeostasis, which facilitates further studies of the physiological functions of iSS.

## Figures and Tables

**Figure 1 antioxidants-11-01292-f001:**
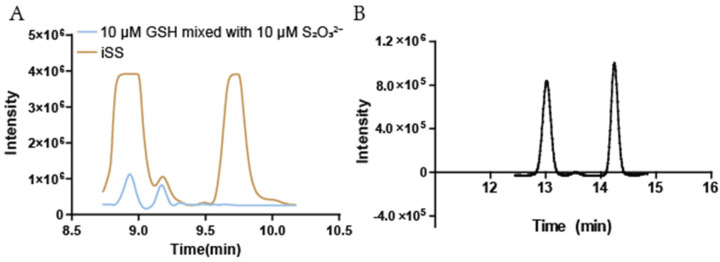
Partial HPLC chromatograms with two programs. The *E. coli* BL21(DE3) was cultured until the OD_600_ reached around 1.0. The cells equivalent to 3 mL at OD_600_ = 1 were collected and the iSS contents were determined with the SdSS method. After derivation with mBBr, the derivatives were determined with two different HPLC programs. (**A**) The previous used HPLC elution. The bimane-S_2_O_3_^2−^ (at 9.2 min) formed a peak overlapped with bimane-GSH (at 8.9 min). The blue curve was the mixture of 10 μM bimane-S_2_O_3_^2−^ and 10 μM bimane-GSH, which was routinely used as standard. The yellow curve was the mBBr-derivatives of an *E. coli* cell lysate. The large peak of bimane-GSH from the cell lysate overlapped with that of bimane-S_2_O_3_^2−^. (**B**) The optimized HPLC elution. The optimized elution separated bimane-GSH (at 10.5 min, not shown) and bimane-S_2_O_3_^2−^ (at 13.0 min).

**Figure 2 antioxidants-11-01292-f002:**
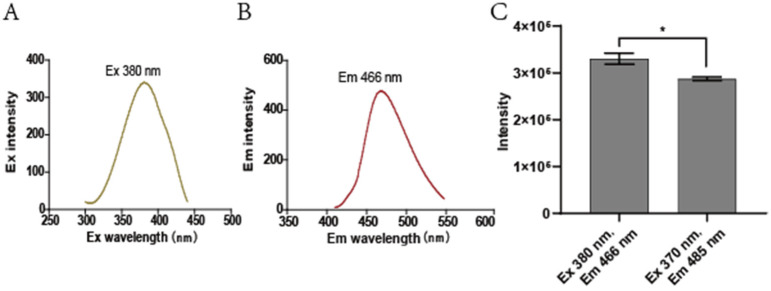
The optimal excitation and emission wavelengths of bimane-S_2_O_3_^2−^. 0.05 mM S_2_O_3_^2−^ was reacted with 0.25 mM mBBr to ensure S_2_O_3_^2−^ was converted to bimane-S_2_O_3_^2−^. This reaction solution was used to search for the optimal excitation and emission wavelengths. (**A**) Fixed Em at 485 nm to screen the Ex from 300 nm to 440 nm. (**B**) Fixed Ex at 370 nm to screen the Em from 410 nm to 550 nm. (**C**) The comparison of the peak area of bimane-S_2_O_3_^2−^ detected with the optimized and original fluorescence detection parameters, respectively. Three parallel experiments were performed to obtain the averages and standard deviations (n = 3). A *t*-test was performed to calculate the *p*-value. Symbol * was shown when *p* < 0.05.

**Figure 3 antioxidants-11-01292-f003:**
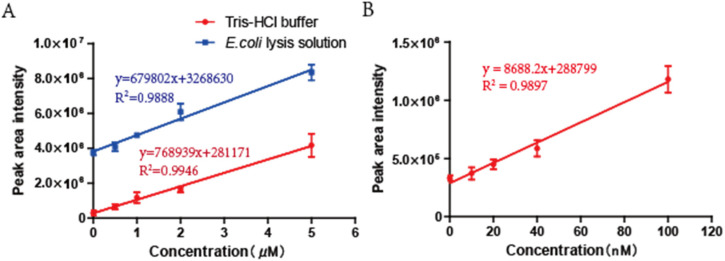
The standard curves of bimane-S_2_O_3_^2−^ range at different concentrations. The standard curves of bimane-S_2_O_3_^2−^ were established ranging from 0 to 5 μM (**A**) and 0 to 0.1 μM (**B**), respectively. The curve in blue color is plotted with cell lysate of *E. coli* BL21(DE3) as a complex background, and the curve in red color is plotted with Tris-HCl buffer as a plain background. Three parallel experiments were performed to obtain the averages and standard deviations (n = 3).

**Figure 4 antioxidants-11-01292-f004:**
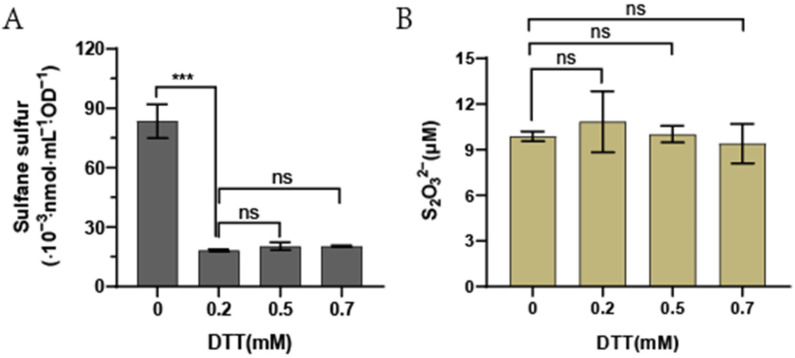
The addition of DTT in the control buffer prevented iSS formation from self-oxidation. (**A**) S_2_O_3_^2−^ detected from 3 mL of *E. coli* BL21(DE3) cells at OD_600_ of 1.0 resuspended in the control buffer with different amounts of DTT. (**B**) DTT did not reduce the amount of S_2_O_3_^2−^. 10 μM S_2_O_3_^2−^ was dissolved in the control buffer with different amounts of DTT and heated at 95 °C for 10 min. The quantity of S_2_O_3_^2−^ was determined after mBBr derivatization. Three parallel experiments were performed to obtain the averages and standard deviations (n = 3). The one-way ANOVA method was performed to calculate the *p*-values (Not significance (ns), *p* ≥ 0.05; ***, *p* < 0.001).

**Figure 5 antioxidants-11-01292-f005:**
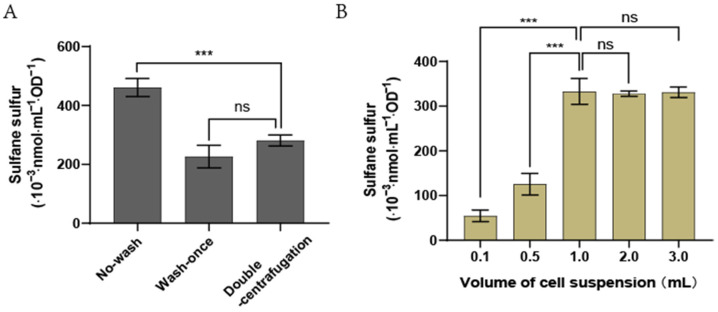
Effects of different treatment conditions on iSS content in *E. coli* cells. (**A**) The harvested cells were treated in three different ways. No-wash, the harvested cells directly proceeded for SdSS analysis without washing; wash-once, the harvested cells were washed once with 50 mM Tris-HCl (pH = 7.4); double-centrifugation, the cell pellets were re-centrifuged and the residual supernatant was removed. (**B**) Different volumes of cell suspension at OD_600_ of 1.0 were harvested and analyzed by using the optimized method. Three parallel experiments were performed to obtain the averages and standard deviations (n = 3). The one-way ANOVA method was used to calculate the *p*-values (ns, *p* ≥ 0.05; ***, *p* < 0.001).

**Figure 6 antioxidants-11-01292-f006:**
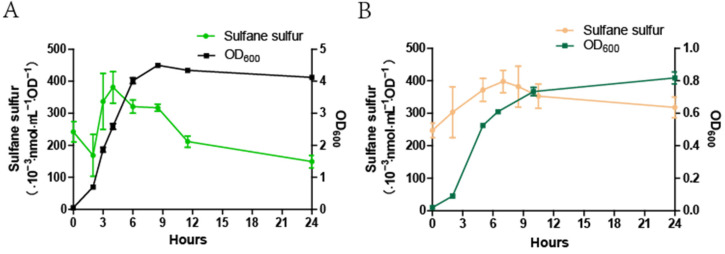
The iSS content of *E. coli* cells cultured under aerobic and anaerobic conditions. The *E. coli* BL21(DE3) cells were incubated in an LB medium and cultured under aerobic (**A**) and anaerobic conditions (**B**). The OD_600_ and iSS content was detected at defined time intervals. Three parallel experiments for each condition were performed to obtain the averages and standard deviations (n = 3).

**Figure 7 antioxidants-11-01292-f007:**
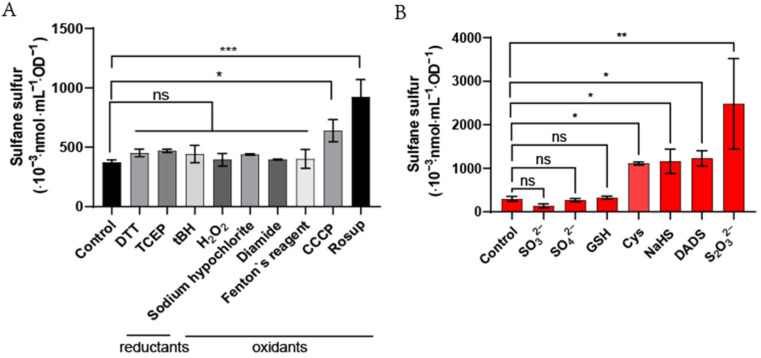
Changes in the iSS content of *E. coli* cells under different oxidants, reductants, and sulfur-bearing compounds. (**A**) *E. coli* BL21(DE3) cells were cultured until their OD_600_ reached 1.0. Then, the cells were aliquoted, and different oxidants and reductants were added and incubated with them for 30 min before iSS detection. The DTT, TCEP, tBH, H_2_O_2_, sodium hypochlorite, diamide, Fenton’s reagent, CCCP, and Rosup were used at 1, 0.2, 0.25, 0.1, 0.1, 1, 0.1 mM, 10 μM, and 50 μg/mL, respectively. (**B**) Different sulfur-bearing compounds at 10 mM were added and incubated with the cell cultures for 30 min before iSS detection. Three parallel experiments were performed to obtain the averages and standard deviations (n = 3). The One-way ANOVA method was performed to calculate the *p*-values (ns, *p* ≥ 0.05; *, *p* < 0.05; **, *p* < 0.01; ***, *p* < 0.001).

**Figure 8 antioxidants-11-01292-f008:**
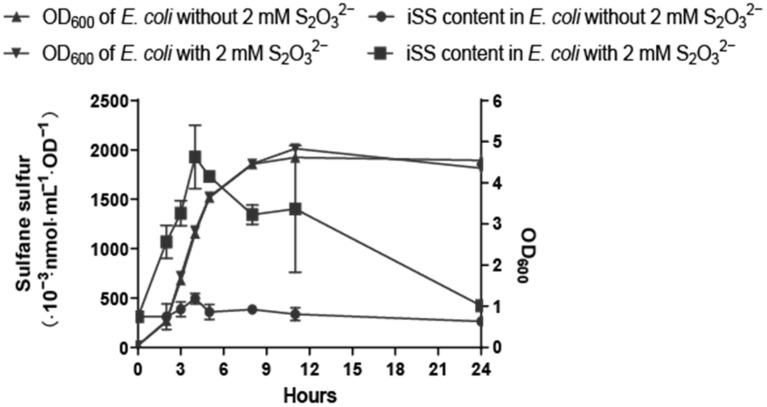
The addition of S_2_O_3_^2−^ increased iSS contents in *E. coli* during growth in LB medium. *E.*
*coli* BL21(DE3) cells were inoculated into fresh LB medium with or without 2 mM S_2_O_3_^2−^ and cultured at 37 °C. The cells at defined time intervals were taken, and the OD_600_ and iSS contents were determined. Three parallel experiments were performed to obtain the averages and standard deviations (n = 3).

## Data Availability

All data are reported in the main text or Appendix A.

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
