# Peer review of "Optimization of a Method for Detecting Intracellular Sulfane Sulfur Levels and Evaluation of Reagents That Affect the Levels in Escherichia coli"

_antioxidants, 2022, doi:10.3390/antiox11071292_

Round 1

Reviewer 1 Report

This superb, fascinating, novel and well documented study concerns the significant field of signaling sulfur molecules in metabolism and mitochondrial function. This study may help to encourage future investigations by scientists in the field of signaling molecules by utilization of improved methods for detection of intracellular sulfane sulfur molecules.

Author Response

Thank you! We appreciate your encouragement and constructive comments and suggestions.

Reviewer 2 Report

This is an interesting article in which authors have reported the an improved and Optimized  method for detection of intracellular sulfane sulfur levels and have evaluated the impact of several oxidizing as well as reducing reagents employing Escherichia coli as a test model. the study is very well designed, experiments well conducted and results are very well described and discussed. a few minor comments are as under:

abstract. E. coli in italics

abstract: S2O32- (typo)

abstract.  tris (2-carboxy....) tris in italics

Introduction and throughout the manuscript: cys full name at first mention and then abbreviation

S-sulfhydration . S in italics

Materials and Methods:  please add the names of cities with countries for both chemicals and equipment.

line 124. please use "the samples were then ..."instead of The samples then were centrifuged....

line 155. remove - between 100 and mL

figure 2 legend. space between number and unit 410 nm, 550, nm

for all figures. please add n=3 when mentioning that experiments were performed three times. 

line 194. "standard curves" instead of stand curves

line 227,233 and Figure 5 both A and B . nmol/mL/OD instead of nmol/ml/OD

 line 303. observe instead of check

line 306. 24 h instead of 24 hours

line 331. typo O2-

discussion. please choose either cysteine residues or cys residues to be consistent

Author Response

Thanks for your kind suggestions! The mentioned formatting and writing errors have been corrected.

Reviewer 3 Report

Review of the paper entitled “Optimization Of A Method For Detecting Intracellular Sulfane Sulfur Levels And Evaluation Of Reagents That Affect The Levels In Escherichia coli” by Qiaoli Yu, Mingxue Ran, Yuqing Yang, Huaiwei Liu, Luying Xun and Yongzhen Xia.

      Sulfane sulfur compounds contain a sulfur atom in the 0 oxidation state covalently bound with another sulfur atom. They are produced in the body through anaerobic cysteine metabolism. Examples of sulfane sulfur-containing compounds include hydropersulfides, polysulfides, polythionates and thiosulfonates. Outer sulfur atom of thiosulfate and elemental sulfur also have these properties. Biological significance of sulfane sulfur consists in its involvement in the regulation of biological activity of proteins by formation of hydropersulfides and trisulfides and in the cyanate detoxification.  Sulfane sulfur-containing compounds show also antioxidant properties.

     However, there is no precise, repeatable and appropriately sensitive method for the quantification of total sulfane sulfur in biological samples. The level of the sulfane sulfur-containing compounds is traditionally determined by the method of Wood [Wood, J.L. Sulfane sulfur. Methods Enzymol. 1987, 143, 25–29] by using a cyanide (CN) which reacts with sulfane sulfur. Reaction product, i.e. thiocyanate (SCN-) and added ferric ions (Fe3+) form colored complex, whose absorbance is measured spectrophotometrically at 460 nm. This method and its modifications, however, are now only rarely used because of the limited sensitivity and specificity [Shinkai Y, Kumagai Y. Sulfane Sulfur in Toxicology: A Novel Defense System Against Electrophilic Stress. Toxicol Sci. 2019 Jul 1;170(1):3-9. doi: 10.1093/toxsci/kfz091. PMID: 30985901]. Thus, a search for new methods of sulfane sulfur detection in biological samples is ongoing. In their previous paper, the Authors presented a new, sensitive method for the quantification of total sulfane sulfur in biological samples [Ran M, Wang T, Shao M, Chen Z, Liu H, Xia Y, Xun L. Sensitive Method for Reliable Quantification of Sulfane Sulfur in Biological Samples. Anal Chem. 2019 Sep 17;91(18):11981-11986. doi: 10.1021/acs.analchem.9b02875. Epub 2019 Sep 4. PMID: 31436086]. In this method, the Authors transform sulfane sulfur present in biological samples into thiosulfate in reaction with sulfite using appropriate conditions of this process (time, temperature, pH). The obtained thiosulfate is converted  by reaction with monobromobimane (mBBr) into thiosulfatebimane which is analyzed via HPLC with a fluorescence detector. The amount of thiosulfate so determined is a measure of the amount of sulfane sulfur in the sample. According to Authors this method is named as sulfite-dependent sulfane sulfur detection method (SdSS).

     In the present paper, the Authors presented the improved SdSS method as they noticed several shortcomings that affect the accuracy and sensitivity of this method. Moreover, based on the improved SdSS method, the Authors investigated the factors that may influence the level of sulfane sulfur in E. coli cells. The Authors emphasize that the key finding of the research carried out is the statement that thiosulfate is a good reagent for increasing sulfane sulfur in E. coli cells, which, however, does not affect the growth of bacteria.

The Authors' proposal is very interesting, however I have some comments.

 My comments

1.      The Authors use two solutions, which they call as: reaction buffer containing sulfite and control buffer without sulfite. This means that the SdSS method only determines the sulfane sulfur pool that has been converted to thiosulfate by reaction with the sulfite, while the thiosulfate content of the samples incubated in the control buffer without sulfite is a kind of blank sample that is subtracted from the test samples. And yet thiosulfate is also a sulfane sulfur-containing compound. I would like the Authors to comment on this fact.

I also don't understand the idea of control buffer containing DTT. DTT lowers the level of sulfane sulfur in the cells (what the Authors also wrote about) because DTT is a reducing agent that converts sulfane sulfur to hydrogen sulfide. So, what is determined in the samples incubated in the control buffer and how is the sulfane sulfur content of the test samples calculated. For what purpose the Authors made this modification. It is not clear. Maybe some graphics (diagram, scheme) showing that the method after the modification is better. All the more so that the Authors point out that “DTT is a key factor for optimization”.

2.      The p-value. I do not understand the information that: not significant (ns) p<0.1234.

Usually a p-value less than 0.05 (typically≤0.05) is statistically significant. There are different p-values in this paper. For example: p< 0.0001; p< 0.0021; p< 0.0332. All these values are less than 0.1234.

3.      „DTT, TCEP, H2O2, tBH, diamide (what kind of chemical compound it is? Reviewer's footnote), and sodium hypochlorite were used at the concentration of 10 mM”. This is an unusually high concentration, and I don't like it. The Authors wrote that „They are the recommended concentrations [37-39]”. It's not true. At number 37. Authors citing the paper of Park et al. (Park YS, Choi SE, Koh HC. PGAM5 regulates PINK1/Parkin-mediated mitophagy via DRP1 in CCCP-induced mitochondrial dysfunction. Toxicol Lett. 2018 Mar 1;284:120-128. doi: 10.1016/j.toxlet.2017.12.004. Epub 2017 Dec 11. PMID: 29241732). Park et al. used micromolar concentrations of CCCP and 1 mM DTT. The Authors also used compounds such as L-cysteine, NaHS, DADS also at a concentration of 10 mM. Sulfur compounds are generally very toxic. Some randomly selected examples from the literature data.

·         The study demonstrates that in Caco-2 and HT-29 cells treated for 6 h, 200 microM DADS increases histone H3 acetylation [Druesne N, Pagniez A, Mayeur C, Thomas M, Cherbuy C, Duée PH, Martel P, Chaumontet C. Diallyl disulfide (DADS) increases histone acetylation and p21(waf1/cip1) expression in human colon tumor cell lines. Carcinogenesis. 2004 Jul;25(7):1227-36. doi: 10.1093/carcin/bgh123. Epub 2004 Feb 19. PMID: 14976134].

·         Various concentrations of DADS ranging from 25 to 100 microM were given to LNCaP cells and the activity of lactate dehydrogenase (LDH) prostatic acid phosphatase (PAcP) and the level of prostate specific antigen were studied [Gunadharini DN, Arunkumar A, Krishnamoorthy G, Muthuvel R, Vijayababu MR, Kanagaraj P, Srinivasan N, Aruldhas MM, Arunakaran J. Antiproliferative effect of diallyl disulfide (DADS) on prostate cancer cell line LNCaP. Cell Biochem Funct. 2006 Sep-Oct;24(5):407-12. doi: 10.1002/cbf.1262. PMID: 16142693].

·         In the permeability and toxicity of L-cysteine in Caco-2 cells studies, the tested cysteine concentrations were: 75, 150, 300, 600, and 1200 µg/mL [Kartal-Hodzic A, Marvola T, Schmitt M, Harju K, Peltoniemi M, Sivén M. Permeability and toxicity characteristics of L-cysteine and 2-methyl-thiazolidine-4-carboxylic acid in Caco-2 cells. Pharm Dev Technol. 2013 Nov-Dec;18(6):1288-93. doi: 10.3109/10837450.2012.659253. Epub 2012 Feb 22. PMID: 22356486].

Apart from that, the Authors did not specify the substances in which they dissolved the tested compounds.

4.      Some figures need to be improved. For example. Figure 7. Concentrations of all compounds should be reported. Preferably in the figure legend. Where the result is statistically significant it should be indicated by an appropriate symbol (*).

5.      “The cell culture without S2O32-were used as the blank control”. I don't understand. I thought blank control was a sample in the buffer from DTT. Besides, where did the Authors get the cell culture without S2O32-?

6.      “This work not only provides a better method for analyzing iSS in biological samples ….”. I think the use of the plural is illegitimate. There is only one kind of biological sample in this paper, namely E. coli bacterial cells. There are no other biological samples.

7.      “S2O32- is the only tested sulfur donor that could increase the iSS without affecting bacterial growth”. Could the Authors comment on this fact further? Why is this the case, because other compounds also supply sulfane sulfur, and only sulfane sulfur supplied by S2O32- does not affect growth. Why? And possibly what is the significance of this fact. Can this information have any practical significance?

8.      The Authors refer the Reader to supplementary materials. But the Authors did not provide any supplementary data. They are not in the system.

 The paper needs to be improved a lot.
